# Presurgical Executive Functioning in Low-Grade Glioma Patients Cannot Be Topographically Mapped

**DOI:** 10.3390/cancers15030807

**Published:** 2023-01-28

**Authors:** Maud J. F. Landers, Lars Smolders, Geert-Jan M. Rutten, Margriet M. Sitskoorn, Emmanuel Mandonnet, Wouter De Baene

**Affiliations:** 1Department of Neurosurgery, Elisabeth-Tweesteden Hospital Tilburg, 5022 GC Tilburg, The Netherlands; 2Department of Cognitive Neuropsychology, Tilburg University, 5037 AB Tilburg, The Netherlands; 3Department of Mathematics and Computer Science, Eindhoven University of Technology, 5600 MB Eindhoven, The Netherlands; 4Hôpitaux de Paris, University of Paris, 75006 Paris, France; 5Service of Neurosurgery, Lariboisière Hospital, 75010 Paris, France

**Keywords:** superior longitudinal fasciculus, frontoparietal network, tractography, executive functions, low-grade glioma

## Abstract

**Simple Summary:**

This study investigated the role of the superior longitudinal fasciculus and frontotemporoparietal network in executive functions. The results demonstrated that neither structural and network overlap nor network disconnection predictors explained executive dysfunction in 156 presurgical IDH-mutated low-grade glioma patients. We specifically looked for features that explain executive dysfunction prior to surgery, both at the level of the individual branches of the superior longitudinal fasciculus (distance to glioma; integrity of tracts) as well as at the network level (via disconnection analyses) with data pooled from two neurosurgical centers. Contrary to our expectations, no predictors were found. We believe it is important to share these null results. It is of interest to neurologists, neurosurgeons, and clinical neuroscientists to know that there is no straightforward topographical explanation of executive dysfunction in presurgical low-grade glioma patients and that we need to develop novel methods to unveil the complex underlying mechanisms. We extensively discuss possible explanations for our findings and suggest how to proceed from here.

**Abstract:**

Executive dysfunctions have a high prevalence in low-grade glioma patients and may be the result of structural disconnections of particular subcortical tracts and/or networks. However, little research has focused on preoperative low-grade glioma patients. The frontotemporoparietal network has been closely linked to executive functions and is substantiated by the superior longitudinal fasciculus. The aim of this study was to investigate their role in executive functions in low-grade glioma patients. Patients from two neurological centers were included with IDH-mutated low-grade gliomas. The sets of preoperative predictors were (i) distance between the tumor and superior longitudinal fasciculus, (ii) structural integrity of the superior longitudinal fasciculus, (iii) overlap between tumor and cortical networks, and (iv) white matter disconnection of the same networks. Linear regression and random forest analyses were performed. The group of 156 patients demonstrated significantly lower performance than normative samples and had a higher prevalence of executive impairments. However, both regression and random forest analyses did not demonstrate significant results, meaning that neither structural, cortical network overlap, nor network disconnection predictors explained executive performance. Overall, our null results indicate that there is no straightforward topographical explanation of executive performance in low-grade glioma patients. We extensively discuss possible explanations, including plasticity-induced network-level equipotentiality. Finally, we stress the need for the development of novel methods to unveil the complex and interacting mechanisms that cause executive deficits in low-grade glioma patients.

## 1. Introduction

The functional architecture of the brain is gradually being disentangled by the study of patients with various types of brain lesions [1,2,3], leading to conclude that focal white matter lesions can cause functional impairments due to disconnections between and within networks [4,5,6,7]. Lesion-symptom studies, in combination with intraoperative stimulation, neuroimaging, and post-mortem dissection studies have established several pathways important for sensorimotor, language, and visual functions that should be spared during surgery to retain the pre-surgical level of functioning [4,5,6,7,8,9,10]. For neurosurgeons, such information is of eminent importance to safely optimize tumor resection while limiting postoperative deficits. This is especially the case in low-grade glioma (LGG) patients as treatment advances have significantly improved median survival of low-grade glioma patients (IDH-mutated oligodendrogliomas and astrocytomas) towards 9–17 years [11]. However, many patients suffer from cognitive disturbances [12] even years after treatment, negatively affecting their daily life [13,14]. Recently, researchers have started looking for pathways involved in executive functions (EF), as it is becoming increasingly clear that cognitive dysfunctions have a high prevalence in low-grade glioma patients and may be the result of structural disconnections of particular subcortical tracts and/or disconnections between or within executive networks [15,16,17,18].

Executive functions can be described as “a set of general-purpose control mechanisms that regulate the dynamics of human cognition and action” [19]. Three core executive functions have been distinguished: inhibitory control, working memory, and cognitive flexibility [20,21]. The frontotemporoparietal network (FTPN) is one of the key networks that has been closely linked to executive functions [22,23,24,25,26,27,28]. The FTPN was previously referred to as the frontoparietal network (FPN), but recent studies indicate that a temporal node is also involved [29,30]. One of the major fiber pathways that substantiate the FTPN is the superior longitudinal fasciculus (SLF) [31,32,33]. The SLF connects the frontal lobe with the temporoparietal junction and the parietal lobe [31,34]. Over the past 40 years, anatomists have been debating about its exact anatomy, studying white matter connectivity patterns in both non-humans and humans, and reached general consensus on three branches belonging to this tract: SLF-I: the superior branch; SLF-II: the more dorsal middle branch; and SLF-III: the more ventral and most lateral branch that partly overlaps with the horizontal segment of the arcuate fasciculus (AF) [35,36,37,38,39,40].

Several studies have associated the SLF with executive functions based on a lesion-symptom mapping approach. These studies found involvement in spatial working memory one month after low-grade glioma surgery for the right SLF I and II [41], in cognitive flexibility four months after surgery in a low-grade glioma case for the right SLF III [42], in cognitive flexibility in the acute and chronic phase following stroke for the left SLF III [43], and in inhibition and cognitive flexibility at least three months after low-grade glioma surgery for the left SLF II and III [18]. Very little research focused on identifying the underlying substrates for EF in preoperative LGG patients, leaving it unclear whether anatomico-functional relationships can be found in this patient group. Potentially, research is limited because the tumoral growing pattern of LGGs induces yet unknown structural and functional changes through neuroplasticity, that complicate the identification of underlying substrates for EF [44]. In healthy individuals, additional evidence of a correlation between SLF and EF can be found in diffusion-weighted imaging (DWI) tractography studies that demonstrated a significant relationship between the structural integrity of the left SLF (without differentiating between branches) and working memory [45]. A recent meta-analysis on fMRI studies combined with tractography results demonstrated the involvement of the SLF-I in a dorsal network related to a spatial motor cluster (amongst others in spatial working memory), the SLF-III in a ventral network related to a non-spatial motor cluster (amongst others verbal working memory and inhibition) and involvement of SLF-II in both networks [46]. Some additional evidence for distinct functions per hemisphere and per branch is found in direct electrical stimulation (DES) studies [47]. Based on DES results, the right SLF-I was associated with a spatial working memory deficit in two patients [41], the right SLF-III was associated with cognitive flexibility in which stimulation led to deficits on the TMT-B in a case report [48], and the left SLF-III was associated with verbal working memory in which stimulation resulted in interferences with the digit span in nine patients [49]. To sum up, no lesion-symptom study has been performed in presurgical low-grade glioma patients that investigated the involvement of the separate branches of the right and left SLF in all three core executive functions (inhibitory control, working memory, and cognitive flexibility).

The aim of this study was to investigate the role of the different branches of the SLF and the FTPN in executive functions in LGG patients from different sets of predictors including (i) spatial proximity of the tumor to the SLF, (ii) structural integrity of the SLF as measured with patient-specific DWI metrics, (iii) overlap between tumor and cortical functional networks including FTPN, and (iv) white matter disconnection measures of the same networks. Spatial proximity and structural integrity of tracts in close vicinity (arcuate fasciculus (AF), corticospinal tract (CST), frontal aslant tract (FAT)), lesion side, tumor grade, and tumor volume were also included in the analyses, to investigate if they account for any of the results. 

## 2. Materials and Methods

### 2.1. Patient Selection

We retrospectively analyzed patients with unilateral frontal, parietal, or temporal low-grade gliomas, grade II and grade III (IDH mutated) who underwent glioma resection between January 2011 and July 2021 in the Elisabeth-TweeSteden Hospital (Tilburg, The Netherlands) or between October 2014 and January 2021 in Lariboisière Hospital (Paris, France). Patients were eligible for the current study if DW-MRI was acquired in the week before surgery and a neuropsychological assessment was conducted in the week before surgery. Exclusion criteria were age below 18 years, a recent history of other major medical illnesses in the year prior to surgery, previous craniotomy, severe psychiatric or neurological disorders in the two years prior to surgery, and lack of basic Dutch or French language skills. This study was approved for both centers by local ethics committees. For the center in Tilburg, the study was approved by the METC Brabant (reference NW2020-32). For the center in Paris, the study was approved by the Comité de Protection des Personnes (CPP), at the Saint-Louis hospital in Paris (reference 2013/51). and all patients gave written informed consent to participate. Patients were informed about the use of their pseudonymized data for the purpose of clinical research and their oral consent was obtained and registered in writing by the clinicians in accordance with the principles of the Declaration of Helsinki.

### 2.2. Measures and Procedure 

#### 2.2.1. DWI Tractography by Means of Constrained Spherical Deconvolution

All DWI scans in the ETZ Tilburg center were acquired using a 3T Philips Achieva MRI scanner (b = 1500, 50 diffusion weighting directions, 6 b = 0 images, 2 mm isotropic voxel size). All DWI scans in the Paris center were acquired using a 3T Siemens Skyra MRI scanner (b = 2000, 64 diffusion weighting directions, 5 b = 0 images, 2.3 mm isotropic voxel size). All scans were transferred to the ETZ center and tractography was performed using the MRTrix 3.0 software package [50]. DWI-MRI data were pre-processed using the MRTrix script *dwifslpreproc*. Probabilistic tractography was performed using the constrained spherical deconvolution-based iFOD2 method with *tckgen*. To seed and restrict tractography of the SLF, estimated regions of interest (ROIs) were identified using SLANT, a method that uses deep learning to generate patient-specific segmentations of 133 anatomical regions [51]. For SLF-I we used the superior frontal gyrus as seed region and the superior parietal lobule and precuneus as target region. For SLF-II we used the caudal half of the middle frontal gyrus and the dorsal half of the precentral gyrus as seed region and the angular gyrus as target region. For SLF-III we used pars triangularis, pars opercularis, and the ventral half of the precentral gyrus as seed region and the supramarginal gyrus as target region. After tracking, spurious streamlines were filtered out of the tracts using fiber-to-bundle coherence [52] as implemented in Dipy [53]. All ROIs were verified by two medical professionals. Tractography for the AF, CST, and FAT was also performed (see tractography protocol) [54].

#### 2.2.2. Tumor Segmentation 

Tumor segmentations were conducted semi-automatically using active contours available in ITK-SNAP [55]. This technique involves some manual assistance to set tumor margins and then automatically segments the tumor. FLAIR images were used to delineate tumor and surrounding tissue. All segmentations were verified by at least two medical professionals. A lesion overlap map is included in Figure 1.

#### 2.2.3. Coregistration and Normalization

For further data processing, all images (T1/T2/Flair, DWI, tractography ROIs, and tumor segmentations) were transferred into each patient’s diffusion-weighted MRI space using NiftyReg affine coregistration [56]. For voxel-wise lesion-symptom mapping (VLSM) and Disconets analyses, tumor segmentations were mapped into MNI space. 

#### 2.2.4. Minimal Distance

The spatial relationship between the ipsilateral tract and the tumor was investigated by calculating the minimal distance. An algorithm calculated the shortest distance between each fiber of the tract and the nearest tumor voxel. The shortest distance over all fibers (in millimeters) was selected as the minimal distance. The potentially strong effect of spurious streamlines on this distance measure was minimized in the filtering step described above.

#### 2.2.5. Structural Integrity

Mean diffusivity (MD) was calculated as a measure of structural integrity based on diffusion tensors calculated using the MRtrix method *dwi2tensor*. MD is an isotropic measure that indicates the magnitude of water diffusion in each direction at a given point, which is inversely related to membrane density. It is a robust marker for pathological processes as it increases due to any disease process that affects barriers [57]. The generated MD values for each voxel of the tract were averaged using *tcksample*, which resulted in average MD. 

#### 2.2.6. Cortical Overlap of the Tumor with Yeo’s Networks

The functional impact of cortical disruption by the tumor was estimated using a parcellation of 17 functional networks described by Yeo et al. [58]. Briefly, this parcellation was obtained by combining resting-state fMRI data of 1000 healthy subjects and detecting clusters of voxels with coherent functional activity. The resulting parcels represent well-known functional networks such as the FTPN. For each of Yeo’s 17 networks [58], the percentage of voxels overlapping the tumor segmentation was calculated, resulting in 17 predictors for each patient. 

#### 2.2.7. Disconets of the Tumor Segmentation

The degree to which the white matter connections within each of Yeo’s 17 networks [58] were disrupted by the tumor was calculated using Disconets [59] (https://github.com/scilus/disconets_flow, accessed on 26 September 2022). Whole-brain tractograms of 20 Human Connectome Project [60] subjects in MNI space were used as representations of healthy brain connectomics. Tractograms were obtained by again using the constrained spherical deconvolution-based iFOD2 method as implemented in MRTrix, generating a set of two million streamlines uniformly seeded in white matter voxels with FOD cutoff set to 0.15. The resulting tractograms were transformed to MNI space and filtered by selecting streamlines with both ends in the same Yeo network. Then, each patient’s tumor segmentation was registered to MNI space with ANTs registration. The disconnection score for each network and for each patient was given by the proportion of streamlines with both ends in that network that intersected the tumor segmentation of that patient, averaged over each of the 20 healthy tractograms. 

#### 2.2.8. Neuropsychological Assessment

In the Tilburg center, neuropsychological assessment was performed between 4 and 1 days prior to surgery and administered as part of standard clinical care. The Dutch version of CNS Vital signs was used, which is a computerized neuropsychological test battery [61]. From this battery, we used two test measures known to recruit executive functions, including the shifting attention test as a measure of cognitive flexibility, and the Stroop interference test (the difference in reaction time on Stroop task 3 − Stroop task 2) as a measure of inhibitory control. In addition, letter fluency was assessed using a Dutch version of the Controlled Oral Word Association Test (COWAT) [62], and working memory was assessed using the digit span test forward and backward [63]. Z-scores were calculated to adjust for age, sex, and education level based on a Dutch normative control sample [64]. Note that z-scores for the letter fluency test were adjusted for education level only, as age and sex were not found to significantly influence test performance [62].

In the Paris center, neuropsychological assessment was performed in the week before surgery using the standard paper-and-pencil version of the digit span forward and backward test [65], the Stroop interference test (the difference in reaction time on Stroop task 3 − Stroop task 2), the Trail Making Test (TMT) and the letter fluency test [66]. Z-scores were computed according to normative data in a reference sample of French healthy people, stratified by age and educational level. Scores were reversed for Stroop interference so that for all tests lower z-scores indicate worse performance.

#### 2.2.9. Voxel-Wise Lesion Symptom Mapping

Standard mass univariate voxel-wise lesion-symptom analyses were conducted using general linear models in NiiStat (http://www.nitrc.org/projects/niistat, accessed on 6 March 2022) with the MNI-registered tumor segmentations and the z-scores of the five EF tests. Only voxels where at least 5% of all patients had tumor overlap were included in the analyses [67]. The alpha value was set at 0.05, one-tailed, as we assumed that tumor overlap would correlate with worse, not better performance. The Benjamini–Hochberg procedure for false discovery rate (FDR) correction was used to control for multiple comparisons for testing multiple voxels and for testing multiple EF tests [68].

### 2.3. Statistical Analysis 

#### 2.3.1. Descriptive Statistics 

Descriptive analyses were performed for the following participant characteristics: age, sex, level of education, affected hemisphere, tumor volume, and mean diffusivity. Baseline characteristics and test performances were compared between centers. Statistical testing included independent samples t-tests or Mann–Whitney U tests (continuous variables, depending on data distribution) and chi-square tests (categorical variables), with significance level of 0.05. 

#### 2.3.2. Degree of Cognitive Dysfunction

The extent to which performances on group level (z-scores) deviated from healthy controls was analyzed with one-sample z-tests with M = 0 and SD = 1. The number of patients displaying performances at a low (−1.5 < Z < −1) or impaired (Z < −1.50) level were counted for each test for each time point to gain insight into the prevalence of clinically relevant dysfunction [69].

#### 2.3.3. Spearman’s Rank-Order Correlation

The relationship between the distance measure, the structural integrity measure and the cognitive outcome measures were explored by conducting a Spearman’s correlation analysis, because variables were not normally distributed. For all analyses, the necessary assumptions for Spearman’s correlation were evaluated. To control for multiple statistical testing, corrected alpha values were calculated and set against *p*-values, using the Benjamini–Hochberg procedure to reduce FDR [68]. False discovery rates were set at 0.1 given the exploratory character of this study [70].

#### 2.3.4. Linear Regression 

To assess whether minimal distance or structural integrity are prognostic factors for executive test performance, linear regressions were run for the significant results from the correlation analyses. Assumptions for linear regression were evaluated. Tumor volume, integrity (MD), and distance (minimal distance) of the other most nearby tracts (for SLF I: CST and SLF II; for SLF II; SLF I, SLF III and FAT; for SLF III: SLF II and FAT) were included in a base model to test in a multivariable model if addition of the distance and/or structural integrity measure to the base model accounted for a significant effect on executive test performance. 

#### 2.3.5. Machine Learning

A random forest algorithm (as implemented in the scikit-learn package https://scikit-learn.org/stable/, accessed on 3 October 2022) was used to predict preoperative executive function as measured by the neuropsychological assessments described above. As input variables we used four sets of predictors as described above: (i) minimal distance of the tumor to each of the ipsilateral SLF branches and the aforementioned control tracts (AF, CST, FAT), (ii) MD of the ipsilateral SLF branches and control tracts as measured with patient-specific DWI, (iii) cortical overlap between the tumor and each of Yeo’s 17 networks [58], and (iv) Disconets of Yeo’s 17 networks. On top of this, we used medical center, lesion side, and tumor volume as covariate predictors in each run, resulting in (i) 9, (ii) 15, (iii) 20, and (iv) 20 predictors, respectively. Note that for distance we only considered the ipsilateral tracts. As targets, we used binary variables indicating for each neuropsychological test described above whether a patient was impaired (Z-score < 1.5) or not impaired (Z-score ≥ 1.5). Classification performance was assessed by calculating the average area under the classification receiver operating characteristic curve (ROC-AUC) for 100 cross-validation splits holding out 25% of the data for performance testing. Each split was stratified to preserve the group proportions in each cross-validation loop. Statistical significance was assessed by permutation testing, randomly shuffling the target variables 1000 times, and recomputing the ROC-AUC for each of the cross-validations splits in the same way as before. To investigate the importance of each feature variable, we used the permutation importance measure provided by the scikit-learn library and reported the variables with the highest scores.

## 3. Results

### 3.1. Descriptive Statistics

In total, 156 patients (100 in Tilburg and 56 in Paris) were included. An overview of the following patient characteristics is provided in Table 1: age, sex, level of education, affected hemisphere, tumor volume, and mean diffusivity of each SLF branch. Comparison of the samples demonstrated a significant difference between centers for a low and middle level of education (*p* < 0.001) and all the mean diffusivity measures of SLF I, II, and III (*p* < 0.001), which was also observed in the nearby tracts (AF, CST, FAT). Groups did not differ significantly on the other patient characteristics nor on the level of baseline cognitive function.

#### 3.1.1. Degree of Cognitive Dysfunction

Table 2 shows the mean scores on the EF tests. Z-tests demonstrated that the patient sample had significantly lower mean performance than the normative sample on the letter fluency in the Tilburg sample (*p* < 0.05), on the Stroop test in the Paris sample (*p* < 0.01), and on the digit span forward and backward test in both centers (*p*’s < 0.01). In the Tilburg sample, the prevalence of low performance ranged from 18% (shifting attention) to 39% (digit span backward) and of impaired performance from 12% (shifting attention) to 20% (digit span backward). In the Paris sample, the prevalence of low performance ranged from 12% (TMT) to 35% (digit span backward) and of impaired performance from 6% (letter fluency) to 16% (digit span forward). So, the proportion of patients with impaired EF performance was higher than expected in the healthy population (6.7% according to the normal distribution) for all EF tests in the Tilburg sample and for all but one of the EF tests (not for letter fluency) in the Paris sample. A previous study from our research group demonstrated that the number of impaired scores in the 103 healthy controls was 5.8% for the digit span forward and 6.7% for the digit span backward, which is significantly lower than the 16% and 20% of impairment, respectively, found in this study [71].

#### 3.1.2. Voxel-Wise Lesion Symptom Mapping

Standard mass univariate VLSM analyses were applied to identify per voxel the relationship between lesion status and performance for any of the EF tests. Analyses were separately performed for each center and for the merged sample. None of the voxels survived multiple comparison corrections at an FDR of 0.05 with and without correcting for tumor volume. 

#### 3.1.3. Correlation Analyses Distance, Structural Integrity, and EF 

Preliminary analyses including visual inspection of scatterplots demonstrated the relationships of both the distance and structural integrity measures with executive test performance to be not perfectly monotonic, but assumptions for Spearman’s rank-order correlation were met. The *p*-values were adjusted to correct for multiple comparisons and are presented in Appendix A. For left SLF I we only found statistically significant positive correlations for distance and the shifting attention test (*rho* = 0.410, *p* < 0.003, *p* < BH-corrected alpha of 0.1) in the Tilburg sample, which was not found for the TMT in the Paris sample. For left SLF II we only found statistically significant positive correlations for distance and the shifting attention test (*rho* = 0.378, *p* < 0.007, *p* < BH-corrected alpha of 0.1), which was not found for the TMT in the Paris sample. For left SLF III we only found statistically significant positive correlations for distance and the TMT test in the Paris sample (*rho* = 0.439, *p* < 0.015, *p* < BH-corrected alpha of 0.1), which was not found for the shifting attention test in the Tilburg sample. For scatterplots of left SLF I, II, and III and cognitive flexibility see Appendix A. When merging the Tilburg and Paris samples for the distance measure, performance on the cognitive flexibility test showed a significant positive correlation with the left SLF I distance (*rho* = 0.328, *p* < 0.003, *p* < BH-corrected alpha of 0.1), the left SLF II distance (*rho* = 0.365, *p* < 0.001) and the left SLF III distance (*rho* = 0.278, *p <* 0.012, *p* < BH-corrected alpha of 0.1). We did not find any statistically significant correlations for structural integrity. 

#### 3.1.4. Linear Regression

Preliminary analyses included a visual inspection of scatter plots and demonstrated a violation of the normality assumption for linear regression for all variables. To deal with this, non-normality bootstrapping (based on 1000 samples) was performed. This generated bootstrapped *p*-values and confidence intervals based on percentiles instead of standard errors, which were used for significance testing. All other assumptions for linear regression were met. Linear regressions were only run for left SLF I, II, and III and cognitive flexibility as correlations for the other tests were not significant. For the Paris sample, minimal distance and structural integrity for left SLF I, II, and III were not significant prognostic factors for performance. For the Tilburg sample, only the distance of left SLF I and left SLF II were statistically significant prognostic factors for performance on cognitive flexibility, whereas structural integrity was not. For SLF I, an extra millimeter in distance led to a 0.042 increase (95%CI 0.012 to 0.056) and for SLF II an extra millimeter in distance led to a 0.038 increase (95%CI 0.010 to 0.074). Due to power-related issues, distance measures of nearby tracts were not included in the model. Merging the Tilburg and Paris samples allowed for correcting for nearby tracts due to increased statistical power. The center was added to the model. For left SLF I, adding distance to the left CST and distance to left SLF II, caused the distance to the left SLF I to no longer be a statistically significant prognostic factor. For left SLF II, adding distance to the left FAT and to left SLF I and III, caused the distance to the left SLF II to no longer be a statistically significant prognostic factor. 

#### 3.1.5. Spatial Proximity 

When applying the random forest model to the merged data using the set of spatial proximity predictors, there was no target variable for which AUC reached statistical significance (all *p*’s > 0.1). The highest AUC was 0.56 (*p* = 0.272) for the letter fluency test, with distance to the right CST and tumor volume being the most important predictors. The same analyses were run on both the Tilburg and Paris sample separately but revealed no significant results.

#### 3.1.6. Structural Integrity 

Again, there was no target variable for which AUC reached statistical significance (all *p*’s > 0.1). The highest AUC was 0.59 (*p* = 0.185) for the backward digit span test, with MD of the left SLF3 and MD of the right AF being the most important predictors. The same analyses were run on both the Tilburg and Paris sample separately but revealed no significant results.

#### 3.1.7. Cortical Parcels-Based Random Forest

As before, there was no target variable for which AUC reached statistical significance (all *p*’s > 0.1 after FDR correction). The highest AUC was 0.64 (*p* = 0.062 before FDR correction) on the letter fluency test, with tumor volume and overlap with Yeo network 6 being the most important predictors. The same analyses were run on both the Tilburg and Paris sample separately but revealed no significant results. 

#### 3.1.8. Disconets-Based Random Forest

For the Disconets approach, there was also no target variable for which AUC reached statistical significance (all *p*’s > 0.1 after FDR correction for testing five cognitive scores). The highest AUC was 0.67 (*p* = 0.038 before FDR correction) on the letter fluency test, with disconnection of Yeo networks 6 and 7 being the most important predictors. The same analyses were run on both the Tilburg and Paris sample separately but revealed no significant results. 

#### 3.1.9. Additional Random Forest Analyses

In order to reduce the risk of overfitting and thereby potentially increasing performance, we performed three additional analyses using the random forest model.

Firstly, we reduced the number of predictors in each set. We removed the predictors derived from the control tracts (AF, CST, FAT) from the structural integrity and spatial proximity sets and implemented the network predictors using Yeo’s 7 networks instead of Yeo’s 17 networks, reducing the number of predictors from 15 to 9 and from 20 to 10, respectively. However, these smaller predictor sets did also not achieve statistically significant classification performance. Secondly, we reformulated the classification problem using three classes of patients instead of two, by including overperforming patients (Z-score > 1.5) as a separate class (as performed in earlier work [48]). These analyses did not achieve better performance either. Thirdly, we reduced the class imbalance by classifying low-performance patients (Z-score < −1) instead of impaired patients (Z < −1.5), which also did not result in significant findings.

Finally, additional analyses including tumor grade (WHO II or III) as a random forest predictor did not result in significant effects.

## 4. Discussion

Our study aimed to explore the role of the different branches of the SLF and the FTPN in the three core executive functions using different sets of predictors in 156 low-grade glioma patients from two different centers. The patient group demonstrated significantly lower mean performance than the normative sample for most of the EF tests and had a higher prevalence of impairments. However, we were unable to find anatomico-functional predictors for executive dysfunction in this group of patients. Significant correlations between cognitive flexibility and distance of the tumor to left SLF-I, II, and III were found, but these did not remain significant when accounting for other nearby tracts. When using a machine learning approach, the random forest analyses demonstrated that neither the structural, nor the cortical network overlap, nor the network disconnection predictors explained executive function performance. Overall, our null results suggest that there is no straightforward topographical explanation of EF performance in LGG patients. 

Patients performed significantly worse on executive function tests than the normative sample of the Dutch population [61] and of the French population [72] for most of the EF tests. Furthermore, the proportion of patients with impaired EF performance was higher than expected for all EF tests in the Tilburg sample and for all but one of the EF tests (not for letter fluency) in the Paris sample. This demonstrates that the inability to find anatomico-functional predictors was not due to the lack of EF deficits as they were clearly present in the LGG group, which was also described in previous studies [17,73].

Very little research focused on the SLF and EF in preoperative LGG patients, which makes it difficult to compare our findings with previous studies. Research that did focus on LGG patients primarily investigated the postoperative course [18], focused on a different task (spatial working memory task [41]), or only focused on one task (TMT) and postoperative change [59], therefore leaving potential predictors for EF deficits in this patient group preoperatively unknown. In fact, to our knowledge, there has only been one study that found preoperative anatomical correlates for cognitive dysfunctions in LGG patients (left IFOF and semantic fluency) [74]. In the current study, we did find a weak predictive value for letter fluency and the disconnection score of Yeo network 6, also called the dorsal attention B network, but this did not remain significant after correction for multiple testing. In stroke patients, anatomic measurements of the left SLF were associated with worse cognitive flexibility in the acute and chronic phases (using the TMT) [43], and in healthy individuals, the left SLF was associated with working memory performance (using tests of the WAIS among which the digit span) and left SLF II and III with verbal working memory and inhibition [45,46].

Theoretically, the lack of statistically significant predictive accuracy in the random forest models can be due to the relatively low proportion of impaired patients. In an absolute sense, there is only a small number of data points for the machine learning model to learn patterns that are predictive of EF impairment. Especially given the number of predictors (9 to 20 predictors per set), the risk of overfitting on such an imbalanced classification problem is present. However, additional analyses in which we made various attempts to reduce the tendency of the model to overfit revealed no significant predictive models either, which suggests that overfitting is not the primary cause of the failure to predict EF impairment. Presumably, the lack of significant findings is due to an absence of any patterns to learn in the data. This explanation is also supported by the lack of significance in the regression analyses. 

Consequently, it seems that EF impairments in LGG cannot be strictly topographically explained, in the sense that we cannot find unique structure–function relationships for a tract/network and EF. EF are widely spread in the brain and rely on interactions between many brain areas [75], but evidence from intraoperative mapping studies suggests that certain hubs or pathways are important for EF [47]. Nevertheless, we were unable to identify these hubs and/or pathways in this patient population. We hypothesize that this is due to the nature of how LGGs impact the brain, which we discuss next. 

LGGs grow slowly and chronically reorganize the brain in ways we do not fully understand yet, which perhaps leaves us with the wrong hypotheses, and a priori lowers the ability to adequately measure the disturbed underlying substrates for EF deficits in this patient category. LGGs, in comparison to fast-growing high-grade gliomas (HGGs), are more likely to induce plasticity and thereby enlarge inter-individual differences in structure–function relationships [76]. This could explain why using (largely) similar methods, we were able to demonstrate EF deficits in a previous lesion-deficit study that also included HGGs [77] since smaller differences in the structure–function relationships in these patients decrease the difficulty of finding lesion-deficit associations. 

Anatomico-functional relationships for EF in LGG have been found in studies that focused on the post-surgical course [18,59]. This might be explained by the acute damage surgery causes locally to the brain, instantly causing functional deficits, as at the periphery of LGGs there is an interface between tumor cells and potential functional tissue that might be acutely disturbed by surgery [78,79]. On the other hand, before treatment LGGs slowly disengage the lesioned area over a longer time period and induce reorganization of several areas, widely distributed over the rest of the brain, slowly and slightly disturbing global functioning. Support for this hypothesis may be found in a neurocomputational framework of Lambon Ralph et al. [80]. That investigated the paradox of modularity vs. equipotentiality in the brain when comparing acute and slow-growing lesions. From a modular perspective, brain damage can independently impair cognitively separable processing steps (modules) that make up complex behaviors, whereas from an equipotentiality view, these steps are fully connected, and brain damage will lead to more distributed impairment but lesser pronounced impairment in the single damaged processing step. In their model, these authors simulated two functions F1 and F2, each relying mainly on two separate subnetworks N1 and N2, albeit with a slight degree of functional overlap (N1 participating very little in F2, and N2 in F1). Acute damage to subnetwork N1 caused major impairment of F1 and left F2 domains almost intact, thus displaying (quasi-)modularity. Conversely, slowly expanding damage to N1, up to its entire destruction, but under continuous training, caused only a small decline in F1. During this extended period, N2 reweighted its connections, effectively taking over F1 from N1, thus demonstrating the equipotentiality of N1 and N2. In this model, the slight decline in F1 arises due to an overloading of the compensating areas in N2. Consequently, whenever a slowly growing lesion is impacting any node of the EF network, a degree of slight impairment should be observed across executive functions since other nodes of this network take over, explaining why we could not correlate deficits with specific lesional topography. It should be observed that no deficit at all should be detected if the tumor would be located in areas far from any node of the EF network, such as in a primary motor or occipital areas. Unfortunately, the rarity of LGG in these regions [81] precludes confirming this hypothesis. In our patient sample, the tumor did not cause any disconnection of Yeo’s fronto-temporo-parietal (control) networks in only 2 out of 156 patients, which explains why we were unable to statistically establish this as a predictor for normal functioning. The previously described discrepancy between LGG and HGG can also be explained in terms of the quasi-modularity/equipotentiality model, as it was demonstrated that tumor growth velocity modulates the functional network topology of remote brain networks, whereby LGG (in comparison to HGG), lead to the lower ability of specialized processing within functionally related brain regions arranged in modules (i.e., lower modularity), but to a higher capacity of the network to rapidly combine and integrate distributed information (i.e., higher integration) [82,83].

### Limitations and Recommendations for Future Research

One of the limitations of this study concerns uncertainties regarding the CSD-based tractography method and its corresponding measures. Any tractography method may generate an unknown number of false positive fibers, which may also influence the accuracy of the corresponding measures. We chose MD to represent structural integrity as it is a well-known robust and often-used measure, but it might not be sensitive enough to detect subtle changes in integrity as it develops in LGGs [84]. However, we did use patient-specific tractography that included a deep learning atlas to identify the regions of interest. Consequently, the tractography results and the corresponding MD measure were presumably more accurate than in most studies that use normalized tractograms instead of patient-specific tractography [85]. Still, research is limited in brain tumor patients investigating structural integrity measures, emphasizing the importance to share our findings. Furthermore, the MD measures for both centers differed, because they depend on the settings of the MRI scanner and of the chosen parameters in the DWI-MRI protocol [86]. Therefore, we were unable to merge the MD measures from both centers in the correlation and regression analyses, leading to lower statistical power. This issue is less problematic for the random forest models, which can “learn” differences in typical MD values for each center. 

Another possible limitation of this study is that the used tests for executive functions slightly differed from each other between centers, with the biggest differences in the tests that were used for cognitive flexibility (SAT in the Tilburg sample, TMT in the Paris sample) and letter fluency (generating words in one minute for three different letters in Tilburg vs. two minutes for one letter in Paris). Furthermore, assessments took place in different centers in different countries. However, when comparing baseline results between the centers we did not find significant differences (apart from education level) and cognitive performances were largely comparable. In addition, when we merged the data, the center did not appear to be a significant predictor of performance, so these slight differences in EF tests most likely did not influence the results.

Finally, the network measures posed some additional limitations. Normalization of scans with severely disrupted anatomy (in our case, due to the effects of the tumor) can be inaccurate, which may have distorted the overlap measures. Furthermore, intrinsic individual heterogeneity in a functional organization can make the use of network atlases in normalized space inaccurate. The same limitations hold for the disconnection analyses, while on top of this, the use of averaged values derived from healthy tractograms may obscure individual differences in structural organization. 

Future research should aim to identify methods that allow for measuring the functional contribution of remote, undamaged regions (also in the contralateral hemisphere) in the search for predictors of EF [87]. This would require measuring the functional organization of the brain, for example using resting-state fMRI (to distinguish an anatomically and functionally intact tract from an anatomically intact but functionally not contributing tract) and task fMRI in the frame of the dynamic network theory, which was recently considered a reliable method to analyze functional brain network changes derived from neuroimaging data [88]. Alternatively, transcranial magnetic stimulation (TMS) may offer insight into functional dependencies in the FTPN [89]. Combining TMS with concurrent fMRI [90], it may be possible to measure the effects of reorganization in individual LGG patients by examining how the observed changes in activity due to stimulation are different from the changes observed in healthy controls. Furthermore, in the future, TMS could be used to map the FTPN in individuals if executive function tests are standardized, as has been demonstrated with motor function mapping [91,92]. Ideally, in order to test the hypothesis of lesion-induced equipotentiality, we would need an imaging modality that would enable us to measure the synaptic weight changes in undamaged areas. The density of glutamate receptors could be a good proxy of such changes, and PET could provide a powerful tool to this end, thanks to recent advances in radioligands [93]. This way, one could study the way in which the brain has reorganized due to the glioma, potentially revealing mechanisms of compensation [94] or cognitive reserve [95]. Another suggestion for future research would be to use patient-specific tractography and functional connectivity in the Disconets analysis. 

We recommend the neurosurgical practice aim for a homogenous neuropsychological assessment of EFs and perform DWI and fMRI in every LGG patient and collaborate in multicenter studies to increase the number of patients to study. This would allow us to study in more detail the hubs and pathways and the interacting networks in relation to the different EFs with higher statistical power and hopefully one day help us to establish the complex underlying mechanisms of EFs and their compensation under slow damage. This in turn could lead to recommendations for surgeons to allow a larger safe resection and an improved chance to avoid introducing postoperative executive dysfunctions.

## 5. Conclusions

Taken together, we demonstrated EF impairments in 156 presurgical low-grade frontal, temporal, or parietal glioma patients from two different centers. The underlying substrates for EF deficits could not be explained using structural predictors of the SLF nor of nearby tracts, nor by network overlap or disconnection predictors of Yeo’s 17 networks. The results indicate that there is no straightforward topographical explanation of executive function performance in LGG patients and suggest plasticity-induced network-level equipotentiality. Furthermore, these findings stress the need for the development of novel methods to unveil the complex and interacting underlying mechanisms that cause EF deficits in LGG patients.

## Figures and Tables

**Figure 1 cancers-15-00807-f001:**
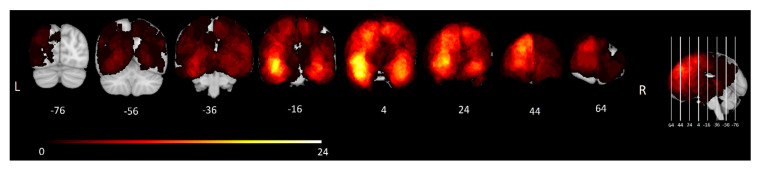
Lesion overlap map. Frequency distribution of tumors in all 156 included patients. The color bar shows the corresponding number of patients with a tumor at each location, ranging from 1 (dark red) to 24 (bright yellow). MNI y-coordinates of each coronal slice are shown below the corresponding slice.

**Table 1 cancers-15-00807-t001:** Patient characteristics (*n* = 156).

		Tilburg T0 (*n* = 100)		Paris T0 (*n* = 56)	
Age mean (SD; range) in years		40.0 (12.0; 19–67)		41.2 (10.8; 27–74)	
Sex (*N*)	Male	62 62.0%		31 55.4%	
	Female	38 38.0%		25 44.6%	
Level of education (*N*) ^1^	Low	21 21.0% *		31 55.4% *	
	Middle	34 34.0% *		5 8.9% *	
	High	45 45.0%		20 35.7%	
Affected hemisphere (*N*)	Right	48 48.0%		39 65.4%	
	Left	52 52.0%		17 34.6%	
Tumor volume median (Q1;Q3) ^2^ in cm^3^	Right	32.66 (2.9; 18.8)		37.06 (6.7; 13.0)	
	Left	49.06 (7.4; 19.0)		43.25 (3.4; 12.8)	
**Mean diffusivity ^1 × 10^−3^^** **Mean (SD)**		**Affected**	**Non Affected**	**Affected**	**Non Affected**
SLF I	Right	0.75 (0.08) *	0.73 (0.06) *	0.64 (0.09) *	0.61 (0.06) *
	Left	0.75 (0.09) *	0.72 (0.06) *	0.61 (0.07) *	0.60 (0.08) *
SLF II	Right	0.73 (0.08) *	0.72 (0.06) *	0.61 (0.09) *	0.60 (0.61) *
	Left	0.74 (0.08) *	0.73 (0.05) *	0.61 (0.73) *	0.59 (0.08) *
SLF III	Right	0.77 (0.1) *	0.76 (0.1) *	0.64 (0.09) *	0.61 (0.07) *
	Left	0.78 (0.1) *	0.76 (0.1) *	0.63 (0.10) *	0.61 (0.09) *

^1^ Education for Tilburg was classified according to Dutch coding system of Verhage and categorized into Low: Verhage 1–4, Middle: Verhage 5 and High: Verhage 6–7; Education for Paris was classified according to the NSC system into Low: 1–3, Middle: 4–5, High: 6–7; ^2^ Quartile 1, median of the lower half of the dataset; Quartile 3, median of the upper half of the dataset; * meaning *p* < 0.05.

**Table 2 cancers-15-00807-t002:** Group-level performances on cognitive tests.

	Center	
	Tilburg T0 (*n* = 100)	Paris T0 (*n* = 56)
Shifting attention/TMT *N*	98	52
mean (SD)	−0.19 (1.08)	−0.11 (1.28)
Low performance *N* (%)	18(18.4)	6(11.5)
of which Impaired *N* (%)	12(12.2)	5(9.6)
Stroop interference *N*	97	52
mean (SD)	−0.13 (1.30)	−0.42 (1.91) **
Low performance *N* (%)	18(18.6)	11(21.2)
of which Impaired *N* (%)	12(12.4)	7(13.5)
Letter fluency *N*	84	53
mean (SD)	−0.27 (1.21) *	−0.02 (1.31)
Low performance *N* (%)	24(28.6)	14(26.4)
of which Impaired *N* (%)	13(15.5)	3(5.8)
Digit Span Forward *N*	46	55
mean (SD)	−0.49 (1.06) **	−0.58 (1.07) **
Low performance *N* (%)	16(34.8)	18(32.7)
of which Impaired *N* (%)	8(17.4)	9(16.4)
Digit Span Backward *N*	46	55
mean (SD)	−0.50 (1.27) **	−0.56 (1.09) **
Low performance *N* (%)	18(39.1)	19(34.5)
of which Impaired *N* (%)	9(19.6)	7(12.7)

** *p* < 0.01; * *p* < 0.05, z-tests comparing patient to controls (M = 0 with SD = 1).

## Data Availability

The data presented in this study are available on request from the corresponding author.

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
