# Peer review of "Presurgical Executive Functioning in Low-Grade Glioma Patients Cannot Be Topographically Mapped"

_cancers, 2023, doi:10.3390/cancers15030807_

Round 1
Reviewer 1 Report
The authors in their article evaluated the role of SLF and FTPN in low grade glioma
Patients , in era of brain mapping and its role in maximum safe resection of glioma with preservation of quality of life after surgery. It is well written and sound study
However I have the following points for the authors to clarify
The level of education for patients in the two centers of the study
Together with site of the lesions whether left or right are different
Without clear information if both affect the results
Authors included grade I to III glioma
Is there any difference according to the grade
It is not clear if they have functional remapping before and after surgery Also if there is any significant difference according to the extent of glioma resection
Author Response
Please see the attachment.
Response to Reviewer 1 Comments
Reviewers response:
The authors in their article evaluated the role of SLF and FTPN in low grade glioma Patients , in era of brain mapping and its role in maximum safe resection of glioma with preservation of quality of life after surgery. It is well written and sound study
However I have the following points for the authors to clarify
Authors response:
First of all, we would like to thank the reviewer for their efforts to review this study. Below we present a point-by-point response to the reviewer’s comments.
Point 1:
The level of education for patients in the two centers of the study
Together with site of the lesions whether left or right are different
Without clear information if both affect the results
Response 1:
The level of education and lesion site were assessed in our study and the results are presented in table 1. As descibed in paragraph 3.1 (p. 7, line 346) level of education differed between the two centers for low and middle level of education (p < .001). However, we calculated z-scores to adjust for eduction level (amongst others) and we did not find any significant effects for any of the groups with any of the analyses (neither with the linear regresion nor with the random forest analyses). Lesion site (left or right) was also assessed and did not differ significantly between centers. In addition, linear regression analyses were performed separately per center and for left and right hemisphere (across centers) depending on the affected hemisphere. Furthermore, in the machine learning analyses, center and lesion side were included as covariate predictors but were not found to influence the results. Therefore, we concluded that both the difference in education level per center and lesion side do not affect the results.
Point 2:
Authors included grade I to III glioma
Is there any difference according to the grade
Response 2:
We only included grade II and grade III IDH mutated low-grade gliomas in this study. We perfomed additional analyses including tumor grade in the random forest model, which did not demonstrate a significant difference between grade II and grade III. To clarify this, we have added the following on p.10, line 469:
Finally, additional analyses including tumor grade (WHO II or III) as random forest predictor did not result in significant effects.
Point 3:
It is not clear if they have functional remapping before and after surgery Also if there is any significant difference according to the extent of glioma resection
Response 3:
The current study focussed on presurgical low-grade glioma patients and only includes presurgical predictors and outcome measures. We agree with the reviewer that it would be interesting to investigate the influence of the extent of the glioma resection at functional remapping, but this is beyond the scope of the current study.

Reviewer 2 Report
To editors and reviewers
Presurgical executive functioning in low-grade glioma patients cannot be topographically mapped
- This is a very interesting manuscript that can be considered for publication in CANCERS. The manuscript is appropriate with aims and scope of journal.
- I suggested some revisions below and after revisions the manuscript can be published.
1) Authors checked and revised all references and citations in your manuscript as MDPI guideline.
2) Add IRB date, number (both M*M and statement) and Helsinki Declaration.
3) Figures are very small. Enhance it.
The content is very good.
Sincerely
Author Response
Please see the attachment.
Response to reviewer 2 Comments
Reviewers response:
Presurgical executive functioning in low-grade glioma patients cannot be topographically mapped
- This is a very interesting manuscript that can be considered for publication in CANCERS. The manuscript is appropriate with aims and scope of journal.
- I suggested some revisions below and after revisions the manuscript can be published.
Authors response:
First of all, we would like to thank the reviewer for their efforts to review this study. Below we present a point-by-point response to the reviewer’s comments.
Point 1:
Authors checked and revised all references and citations in your manuscript as MDPI guideline.
Response 1:
We have checked our references and citations and have revised them to comply with the MDPI guidelines.
Point 2:
Add IRB date, number (both M*M and statement) and Helsinki Declaration.
Response 2:
The date and number of the ethical approval was listed in section 2.1, and holds for both centers. We have clarified this in p. 4, line 177:
This study was approved for both centers by the local ethics committee (NW2020-32, METC Brabant, The Netherlands) and all patients gave written informed consent to par-ticipate.
We have added a statement that our procedures comply with the principles of the Declaration of Helsinki to the manuscript in p. 4, line 184:
Patients were informed about the use of their pseudonymized data for the purpose of clinical research and their oral consent was obtained and registered in writing by the clinicians in accordance with the principles of the Declaration of Helsinki.
Point 3:
Figures are very small. Enhance it.
Response 3:
We have increased the resolution of the figure embedded in the manuscript.

Reviewer 3 Report
The authors present a retrospective study on 156 patients wiht LGG and affection of the frontotemporoparietal network. Special attention was paid to the role of the superior longitudinal fasciculus. Thereby, there was a significantly lower mean performance than the normative sample on most of the executive function tests and had a higher prevalence of impairments. However, both the regression and random forest analyses did not demonstrate significant results, meaning that neither the structural, nor the cortical network overlap, nor the network disconnection predictors explained executive function performance in presurgical low-grade glioma patients. The results revealed that there is no straightforward topographical explanation of executive function performance in LGG patients and suggest plasticity-induced network-level equipotentiality as the authors summarized finally.
The manuscript is well written and the methods are good. The topic is of high interest and the analysis is well performed. However, there are several aspects which should be considered before the manuscript migth be accepted for publication to improce the content:
1. the abstract should be more focused and the main messages must be explored
2. there is a high amount if literature about transcranial magnetic stimulation (TMS) or intraoperative mapping and the tractrographie, identification of networks. These references should be included and discussed in correlation to the authors´different approach.
Functional connectivity of the frontotemporal network in preattentive detection of abstract changes: Perturbs and observes with transcranial magnetic stimulation and event-related optical signal.
Xiao XZ, Shum YH, Lui TK, Wang Y, Cheung AT, Chu WCW, Neggers SFW, Chan SS, Tse CY.Hum Brain Mapp. 2020 Aug 1;41(11):2883-2897. doi: 10.1002/hbm.24984. Epub 2020 Mar 14 Establishing the functional connectivity of the frontotemporal network in pre-attentive change detection with Transcranial Magnetic Stimulation and event-related optical signal. Tse CY, Yip LY, Lui TK, Xiao XZ, Wang Y, Chu WCW, Parks NA, Chan SS, Neggers SFW.Neuroimage. 2018 Oct 1;179:403-413. doi: 10.1016/j.neuroimage.2018.06.053. Epub 2018 Jun 19. A guide for concurrent TMS-fMRI to investigate functional brain networks. Riddle J, Scimeca JM, Pagnotta MF, Inglis B, Sheltraw D, Muse-Fisher C, D'Esposito M.Front Hum Neurosci. 2022 Dec 15;16:1050605. doi: 10.3389/fnhum.2022.1050605. eCollection 2022. Mapping of Motor Function with Neuronavigated Transcranial Magnetic Stimulation: A Review on Clinical Application in Brain Tumors and Methods for Ensuring Feasible Accuracy. Sollmann N, Krieg SM, Säisänen L, Julkunen P.Brain Sci. 2021 Jul 7;11(7):897. doi: 10.3390/brainsci11070897. Preoperative navigated transcranial magnetic stimulation and tractography in transparietal approach to the trigone of the lateral ventricle. Hendrix P, Senger S, Griessenauer CJ, Simgen A, Linsler S, Oertel J.
Preoperative Navigated Transcranial Magnetic Stimulation Improves Gross Total Resection Rates in Patients with Motor-Eloquent High-Grade Gliomas: A Matched Cohort Study.
Hendrix P, Dzierma Y, Burkhardt BW, Simgen A, Wagenpfeil G, Griessenauer CJ, Senger S, Oertel J.Neurosurgery. 2021 Feb 16;88(3):627-636. doi: 10.1093/neuros/nyaa486 3. The authors should be perform this study prospectively. Why selected they a retrospective analysis? This might improve the study clearly. 4. It would be helpful for the understanding, that the authors present some imaging of the selected LGG patients to get an idea of the approach and treatment options. 5. Do the authors have a key message for treatment based on their results? Should this kind of mapping and testing introduced into the clinical daily routine for the surgical tretament?Author Response
Please see the attachment.
Response to reviewer 3 Comments
Reviewers response:
The manuscript is well written and the methods are good. The topic is of high interest and the analysis is well performed. However, there are several aspects which should be considered before the manuscript migth be accepted for publication to improce the content:
Authors response:
First of all, we would like to thank the reviewer for the efforts to review this study. Below we present a point-by-point response to the reviewer’s comments.
Point 1:
The abstract should be more focused and the main messages must be explored
Response 1:
We agree that the abstract was too long and have shortened it, thus improving the focus on the main messages.
Point 2:
There is a high amount if literature about transcranial magnetic stimulation (TMS) or intraoperative mapping and the tractrographie, identification of networks. These references should be included and discussed in correlation to the authors´different approach.
Functional connectivity of the frontotemporal network in preattentive detection of abstract changes: Perturbs and observes with transcranial magnetic stimulation and event-related optical signal.
Response 2:
TMS could indeed be used as an additional tool to investigate functional reorganization or to map networks important for executive functions in individuals. We have added a discussion of these ideas and have incorporated some of the suggested literature in p.12 starting from line 616:
Alternatively, Transcranial Magnetic Stimulation (TMS) may offer insight into functional dependencies in the FTPN. Combining TMS with concurrent fMRI, it may be possible to measure the effects of reorganization in individual LGG patients by examining how the observed changes in activity due to stimulation are different from the changes observed in healthy controls. Furthermore, in the future, TMS could be used to map the FTPN in individuals if executive function tests are standardized, as has been demonstrated with motor function mapping.
Point 3:
The authors should be perform this study prospectively. Why selected they a retrospective analysis? This might improve the study clearly.
Response 3:
Since we only focussed on preoperative predictors and preoperative executive functioning, our analyses did not depend on decisions made or tools used during treatment of patients. Furthermore, All patients included in this study had been selected for tumor resection via craniotomy by the respective surgical teams, either via an awake procedure or under general anesthesia, after data collection. Therefore, we feel that the retrospective nature of our study did not limit our ability to draw conclusions from our data.
Point 4:
It would be helpful for the understanding, that the authors present some imaging of the selected LGG patients to get an idea of the approach and treatment options.
Response 4:
Since we only focussed on preoperative predictors and outcomes, our data and analyses did not depend on the choice of treatment options or on treatment outcomes. We exclusively sought to uncover the mechanisms by which LGGs cause preoperative executive dysfunctions. Therefore, we feel that studying treatment options and their effects is outside the scope of this study.
Point 5
Do the authors have a key message for treatment based on their results? Should this kind of mapping and testing introduced into the clinical daily routine for the surgical tretament?
Response 5:
Unfortunately, our results do not allow us to suggest improvements to treatment, since we did not find any significant predictors of preoperative executive function. If, as we suggested, novel predictive models are able to predict executive dysfunction in low-grade glioma patients, we may learn from these models how executive (dys)functions arise from brain structure. This in turn could lead to recommendations for surgeons to allow a larger safe resection and an improved chance to avoid introducing postoperative executive dysfunctions. We discuss this in paragraph 4.1 starting from line 630 and have added the latter sentence for further clarification (line 634).

Round 2
Reviewer 3 Report
The authors present their revised manuscript. They answered to reviewer´s comments adequately. Acceptable as is it.